# Oxidative Stress in Antibiotic Toxic Optic Neuropathy Mimicking Acute LHON in a Patient with Exacerbation of Cystic Fibrosis

**Lea Kovač [1], Marija Volk [2], Maja Šuštar Habjan [1] and Marko Hawlina [1,\*]**

[1]   Eye Hospital, University Medical Centre Ljubljana, 1000 Ljubljana, Slovenia
[2]   Clinical Institute of Genomic Medicine, University Medical Centre Ljubljana, 1000 Ljubljana, Slovenia
\*   Correspondence: marko.hawlina@kclj.si

**Abstract:** The striking similarity of disc edema without leakage on fluorescein angiography, which is pathognomonic of Leber hereditary optic neuropathy (LHON), was present in a patient with cystic fibrosis with antibiotic toxic optic neuropathy. This similarity suggested the common effect of oxidative stress on retinal ganglion cells in inherited mitochondrial and antibiotic optic neuropathies. We present the case of a patient with advanced cystic fibrosis on chronic antibiotic treatment who experienced a rapid painless bilateral visual decline over a course of a few weeks. At examination, his corrected visual acuity was reduced to 0.3 in both eyes, with dyschromatopsia and central scotoma. The appearance of the fundus resembled the typical clinical features of acute LHON with hyperemic optic discs and tortuous vessels with no dye leakage from the optic discs on fluorescein angiography. Ganglion cell layer loss was seen on optic coherence tomography, with all findings pointing to LHON. Genetic testing did not reveal any LHON-specific mutations. After extended genetic testing, a heterozygous variant c.209C>T in the *OPA3* gene on chromosome 19, g.46032648G>A, classified as a variant of unknown significance, was also found. After discontinuing antibiotics and general improvements in his health, surprisingly, his visual function completely improved. Later, he also received a bilateral lung transplant that further improved his general condition, and his vision remained normal. Excluding LHON, the transient optic neuropathy in our patient could be mainly due to antibiotic toxicity of linezolid and ciprofloxacin, which have been linked to mitochondrial dysfunction and advanced cystic fibrosis with hypoxic status. We suggest the possibility that patients with cystic fibrosis may be more prone to developing mitochondrial optic neuropathy, especially with additional risk factors such as chronic antibiotic therapy, which affect mitochondrial function, and can perhaps serve as a model for LHON.

**Keywords:** mitochondrial optic neuropathy; *OPA3*; cystic fibrosis; antibiotic toxicity; case report

## 1. Introduction

Mitochondrial optic neuropathies are a diverse subgroup of optic nerve disorders [1]. Classical prototypes of mitochondrial neuropathies are Leber hereditary optic neuropathy (LHON), which is caused by mutations in the mitochondrial DNA [1] or the nuclear recessive *DNAJC30* gene [2], and dominant optic atrophy (DOA) secondary to mutations in the nuclear *OPA1* genes that encode mitochondrial inner membrane proteins [1]. LHON most commonly affects young males and presents as painless, acute, or subacute bilateral central loss of vision (consecutive or simultaneous). Although the appearance of the fundus may be normal at the time of the visual loss, many patients show optic disc hyperemia, edema of the peripapillary retinal nerve fiber layer without dye leakage on fluorescein angiography (pseudoedema), retinal telangiectasia and increased vascular tortuosity [3]. The loss of retinal ganglion cells (RGCs) is a neuropathological feature [4], predominantly affecting RGCs within the papillomacular bundle. Conversely, mitochondrial damage resulting in bilateral optic neuropathy is more insidious in DOA, with variably affected and

slowly progressive loss of visual acuity without disc edema. Environmental factors, such as various toxins, medication and nutritional deficiencies, can also affect mitochondrial metabolism in the optic nerve [1]; however, clinically mimicking acute LHON was achieved with systemic antibiotics that affect oxidative phosphorylation in the respiratory chain in the mitochondria, resulting in an acute increase in and accumulation of reactive oxygen species (ROS), which triggered a proapoptotic signaling cascade. We present a case of toxic optic neuropathy mimicking acute LHON, suggesting a common mechanism of oxidative stress causing acute mitochondrial dysfunction, that improved with the cessation of toxic antibiotics and an improvement in the general health and oxygenation.

## 2. Case Description

A 28-year-old male presented to our clinic with gradual worsening of visual acuity in both eyes, accompanied by worsening of his hearing and paraesthesias.

He had experienced a recent exacerbation of long-standing cystic fibrosis that had reached a terminal stage, requiring a lung transplant, accompanied by secondary diabetes mellitus, osteoporosis, liver cirrhosis, nephrolithiasis, pseudomonas aeruginosa and MRSA colonization. Due to secondary chronic pulmonary infections and other complications of his primary disease, he had been receiving several antibiotics for at least a year (azithromycin, 500 mg once every 2 days; ciprofloxacin, 750 mg twice daily; tobramycin, 300 mg/4 mL by inhalation twice daily; linezolid, 600 mg twice daily), as well as dexamethasone (0.5 mg), pantoprazole, insulin, enzyme supplements, ursodeoxycholic acid, iron supplements, calcium carbonate, 7% NaCl and deoxyribonuclease inhalation. There was no family history of systemic or ocular disease.

On examination at the first presentation, his visual acuity was 0.5, and it further declined to 0.3 in both eyes over one week. The Ishihara color plate scores were 9/21 in the right eye (RE) and 13/21 in the left eye (LE). The pupillary reflexes were preserved, and no relative afferent pupillary defect was detected, as both eyes were similarly affected. Automated perimetry showed central scotomas in both eyes. A biomicroscopic examination of anterior parts of the eyes was normal. In both fundi, the optic discs appeared hyperemic and swollen, and the vessels tortuous and distended (Figure 1A).

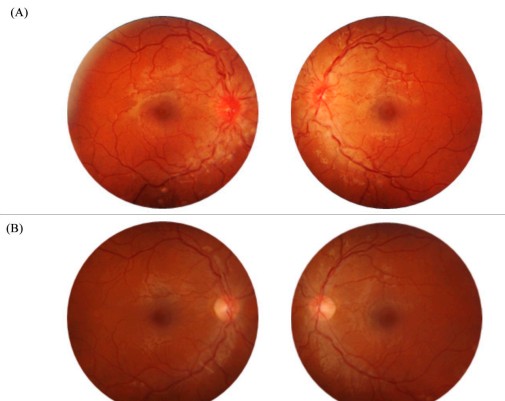

(A)

(B)

**Figure 1.** Right and left fundus on presentation, showing the hyperemic optic disc and tortuous vessels (**A**) and a normal fundus appearance after five months (**B**).

Macular Optic coherence tomography (OCT) showed thickening of the disc's retinal nerve fiber layer (RNFL) and thinning of the ganglion cell layer (GCL) (Figure 2A). Fluorescein angiography showed no dye leakage in the optic discs (Figure 3). Electrophysiology testing showed a preserved pattern electroretinogram (PERG) (Figure 4) and significantly reduced the visual evoked potentials (VEP) in both eyes. The full blood tests, including inflammatory markers, was normal. Serology tests for *Borrelia burgdorferi*, *Treponema pallidum* and *Mycoplasma pneumoniae*, and the quantiferon test were negative. Herpes virus and varicella zoster virus showed only positive IgG antibodies but negative IgM antibodies, pointing to an older exposure. The extractable nuclear antigen (ENA) panel, antinuclear antibodies

(ANA) and anti-neutrophil cytoplasmic antibodies (ANCAs) were not detected. The serum angiotensin-converting enzyme level was normal. Serum levels of Vitamin E, Vitamin B12 and folic acid were within the normal range. In the head and cervical spine, magnetic resonance FLAIR, STIR-FLAIR, T1-weighted, T2-weighted, diffusion-weighted and susceptibility-weighted imaging were performed via a standard multiple sclerosis protocol (Figure 5). Since there were no apparent pathological changes along the optic nerves, intracranially or along the cervical spine, the gadolinium contrast dye was not applied. An examination by an ear–nose–throat specialist showed a normal tympanogram and audiogram and normal findings. A neurological examination revealed possible peripheral sensory polyneuropathy of the gloves/socks type. Sensory polyneuropathy was later confirmed by an electromyogram.

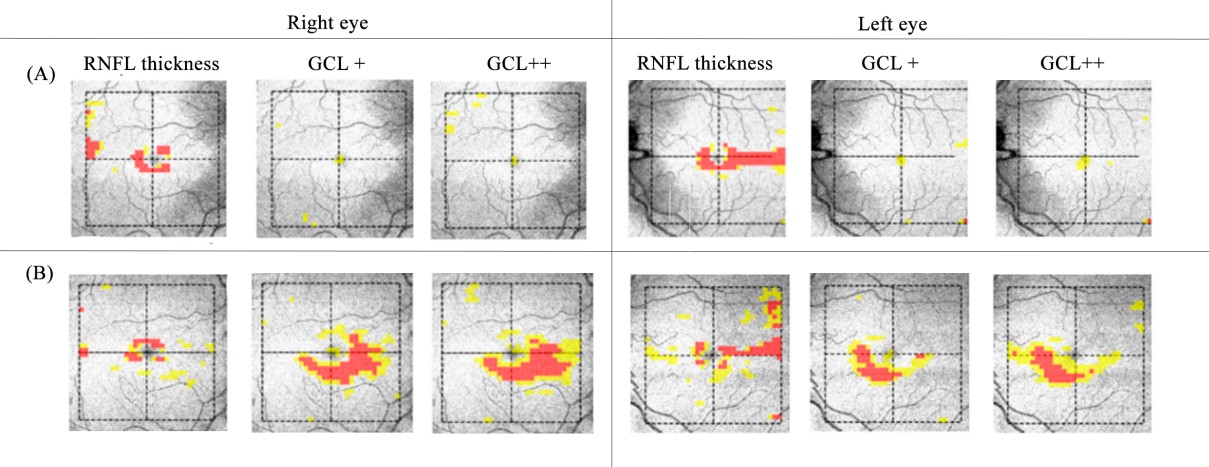

**Figure 2.** Retinal ganglion cell layer thinning revealed by optic coherence tomography (Topcon) at presentation (**A**) and after five months (**B**).

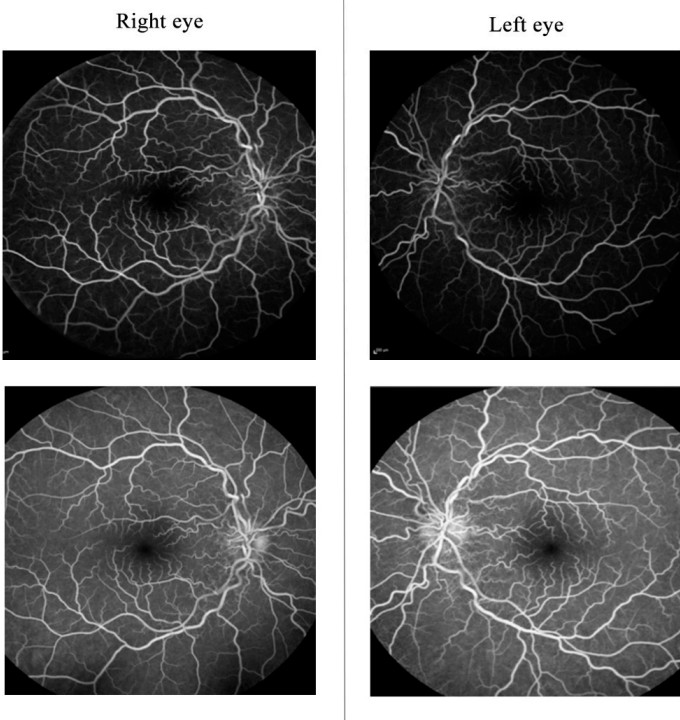

**Figure 3.** Fluorescein angiography showing tortuous bilateral telangiectatic vessels and late disc staining with no dye leakage.

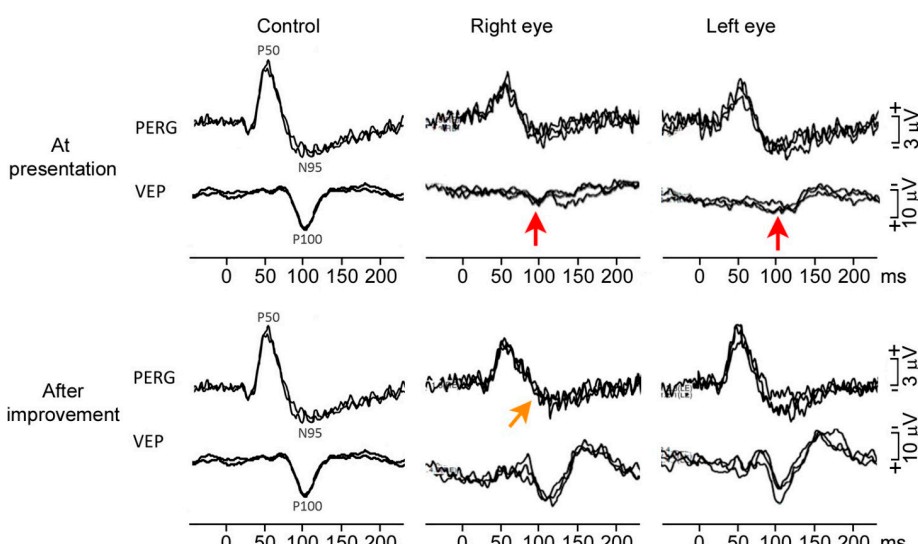

**Figure 4.** The P100 wave of the visual evoked potentials was diminished before treatment (red arrows) and normalized after treatment. A slightly reduced PERG N95 wave in the right eye after five months (orange arrow), showing slight retinal ganglion cell damage in the right eye and normal findings in the left eye.

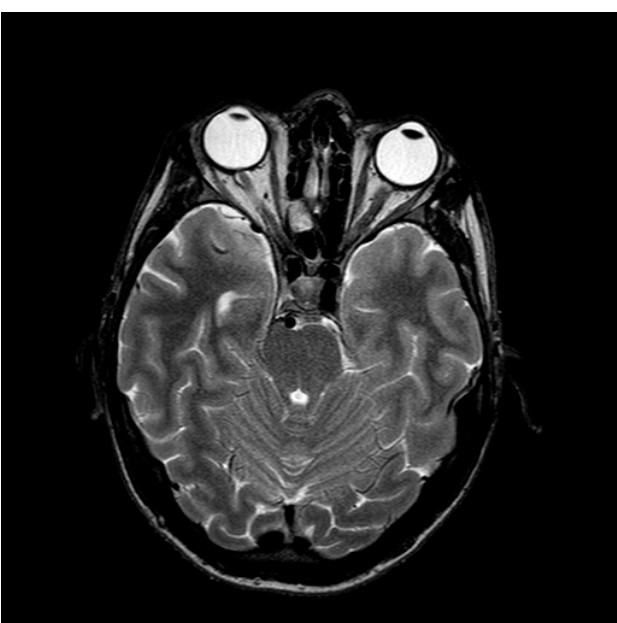

**Figure 5.** Magnetic resonance imaging of the head, showing no pathological changes intracranially or in the course of the optic nerves.

Having excluded other causes of bilateral optic neuropathy, our primary differential diagnosis was Leber hereditary optic neuropathy (LHON). Genetic testing for hereditary optic neuropathies, including sequencing the whole mitochondrial genome and exome, did not reveal any typical or atypical mutations for LHON. However, a novel heterozygous variant c.209C>T of the optic atrophy 3 (*OPA3*) gene on chromosome 19, g.46032648G>A, was found, which was classified as a variant of unknown significance. Segregation analysis was advised, but by the time of submitting this manuscript, the patient's parents had not responded to the extended invitation for genetic testing, since no other family member had any visual problems.

The patient's antibiotic treatment with ciprofloxacin and linezolid was discontinued, and he received supportive therapy in form of supplemental vitamins (B1, B12, B6 and

biotin) and idebenone (900 mg daily). He reported a gradual improvement in his visual acuity within the first month. After two months, he underwent a previously scheduled lung transplant, which contributed to a general improvement in his health, and was closely followed by pulmologists.

In a follow-up ophthalmic examination after 6 months, his visual acuity had returned to 1.0 bilaterally, his color vision was full, and his visual field had improved significantly with only mild paracentral scotoma remaining. The appearance of the fundus returned to normal (Figure 1B), though a moderate loss of ganglion cells was observed by macular OCT (Figure 2B). Repeated electrophysiological tests (Figure 4) showed a slightly reduced PERG N95 wave of the RE in accordance with the partial loss of ganglion cells, whereas the ERG pattern of the LE was still within normal limits. VEP signals from both eyes improved significantly (normal to full-field pattern stimulation and only borderline abnormal to half-field stimulation). An outline of clinical course is resumed in Table 1.

**Table 1.** Timeline of the clinical course.

| Timeline | Clinical Progression |
| --- | --- |
| ≥1 year prior to the symptoms' onset | Combination antibiotic therapy with linezolid, azithromycin, ciprofloxacin, and tobramycin |
| May/June 2020 | Hearing worsening; paresthesia in the hands, stomach and feet |
| July 2020 | Blurry vision in both eyes |
| 24 July–3 August 2020: hospitalization | At admission: BCVA RE, 0.5; LE, 0.5. At discharge: BCVA RE, 0.3; LE, 0.3 Ishihara color plate score: RE, 9/21; LE, 13/21; no relative afferent pupillary defect; central scotomas in both eyes Hyperemic and swollen optic discs, tortuous and distended vessels Diagnostic tests: full blood panel, serology, head and cervical spine MRI, genetic testing, ENT examination, neurological, electrophysiology Discontinued ciprofloxacin and linezolid, started on supplemental therapy with biotin; Vitamins B1, B6, and B12; and idebenone |
| September 2020 | Subjective improvement in visual acuity (as subsequently reported by the patient) |
| October 2020 | Lung transplant |
| 18 January 2021: first outpatient follow-up | BCVA RE, 0.9; LE, 0.8 Ishihara plates: full Kinetic Goldmann perimetry: no scotoma Fundi normal, no visible disc edema |
| 4 February 2021 | Repeat electrophysiology: a slightly reduced PERG N95 wave in the RE; normal in the LE; significantly improved VEP signals from both eyes Macular OCT: GCL thinning in both eyes |
| 22 March 2021: second outpatient follow-up | BCVA RE, 1.0; LE, 1.0 Ishihara plates: full Automated static perimetry (Octopus, G2 TOP strategy) within normal limits |

## 3. Discussion

The case we report here is interesting, as it associated a LHON-like phenotype with cystic fibrosis and chronic use of antibiotics. It is assumed that it indicates a mitochondrial optic neuropathy with reversible visual loss, which initially presented in a similar way to LHON, but was likely precipitated by antibiotics in a predisposed individual with an additional genetic disorder, which possibly also affected his mitochondrial function.

The genetic variant in the *OPA3* gene in our patient has not been previously identified; however, it is known that *OPA3* genes encode a protein in the mitochondrial outer membrane with pro-fission properties, and disturbed mitochondrial dynamics have been linked to the loss of RGCs [5]. Furthermore, known heterozygous mutations in *OPA3* genes have been associated with early-onset bilateral optic atrophy and cataracts, while mutations in

both *OPA3* alleles cause autosomal recessive 3-methylglutaconic aciduria Type III with later-onset optic atrophy, extrapyramidal dysfunction, spasticity and ataxia [6–10]. Additional features have been described, such as peripheral neuropathy [10,11] and hearing loss due to the expression of *OPA3* in murine cochlear tissue [11,12], and it may be of interest that our patient experienced hearing loss, which is not typical in LHON. Available data on the epidemiology of *OPA3* mutations in the general population are scarce, since genetic studies have focused predominantly on symptomatic individuals or their relatives [6,10,11,13–17]. Furthermore, patients with LHON-like mitochondrial neuropathies, even without primary LHON mutations, may express mitochondrial disturbances that likely contribute to the vulnerability of the optic nerve [18]. As the visual loss in our patient presented with disc pseudoedema and was reversible, which would not be expected for the genetic variant in the *OPA3* gene, this variant might thus play a role in increased susceptibility for mitochondrial damage.

Secondly, our patient's primary genetic disease, cystic fibrosis (CF), could itself be a factor influencing mitochondrial activity. CF is a genetic autosomal recessive multiorgan disease due to mutations in the cystic fibrosis transmembrane conductance regulator (CFTR) gene, which are traditionally known to cause the dysfunction of a transmembrane chloride channel, resulting in changes in the biological fluid and electrolyte homeostasis [19]. Recent research, however, has suggested that there are other cellular functions affected by this mutation: reduced CFTR protein activity was shown to downregulate the expression of the mitochondrial *ND4* gene, thus diminishing the activity of mitochondrial Complex I [20]. The same mitochondrial *ND4* gene is affected in one of the three typical LHON mutations, m.11778G>A [3]. Optic neuropathies have previously been described in patients with CF, presenting with similar clinical findings as were seen in our patient (hyperemic disc edema, tortuous vessels, flame-shaped hemorrhages, cystic intraretinal changes, central scotomas) [21]. While the connection to mitochondrial damage was not yet known at that time, the optic neuropathies seemed to have been linked to severity of the pulmonary disease, since the improvement in visual function coincided with an improvement in his general health condition [21]. Other cases of optic neuropathies in patients with CF have been attributed to the toxicity of the prolonged systemic chloramphenicol treatment that was commonly used as treatment for children with chronic pulmonary infections in the 1950s and 1960s [22]. Later, it was found that chloramphenicol inhibits mitochondrial protein synthesis, causing a change in the mitochondrial ultrastructure and a decrease in the production of ATP. This leads to acute to subacute bilateral painful or painless vision loss, cecocentral scotomas, optic disc edema, retinal vessel tortuosity and retinal hemorrhages [23].

Other antibiotics, such as macrolides (erythromycin, ciprofloxacin) and linezolid, can also cause disruptions in mitochondrial function [23,24]. Linezolid was shown to inhibit the synthesis of ATP in the mitochondria, which decreased the synthesis of fatty acid [25], interfered with the ribosomes, disrupted mitochondrial oxidative phosphorylation and inhibited the synthesis of mtDNA-encoded subunits of proteins in respiratory chain complexes [26]. This resulted in significantly decreased activity of the mitochondrial respiratory chain Complexes I and IV [26]. Several reports have linked long-term linezolid toxicity with optic neuropathy that presented as bilateral hyperemic disc edema, vessel tortuosity and no dye leakage on fluorescein angiography, and improved upon discontinuing linezolid [27–36], as it did in our patient. As Javaheri et al. [27] suggested, the mechanism of linezolid's influence on mitochondrial function can theoretically mimic the respiratory chain dysfunction in LHON, where respiratory chain Complex I is most commonly affected. Any disruptions in oxidative phosphorylation in the respiratory chain in the mitochondria result in a significant increase in and accumulation of reactive oxygen species (ROS), which trigger a proapoptotic signaling cascade [37] and may also trigger compensating increases in the mitochondria, manifesting as retinal nerve fiber edema [38]. In its acute phase, LHON clinically presents in a similar fashion to linezolid-induced optic neuropathies, with telangiectatic microangiopathy, vessel tortuosity, optic disc swelling and hyperemia, but

no leakage noted in fluorescein angiography [39–42]. The unmyelinated prelaminar and intraocular portion of RGC axons represent sites with high energy demands [43], rendering the optic nerve particularly sensitive to mitochondrial dysfunctions. Due to the considerably smaller caliber of RGCs and the limited axoplasmic transport of the mitochondria, neurons in the papillomacular bundle might be more susceptible to mitochondrial damage compared with other RGCs [39].

A striking common feature of mitochondrial optic neuropathies is the relatively selective degeneration of the papillomacular bundle of RGCs, suggesting a common pathophysiological pathway [37]. In reported cases of linezolid-induced toxic optic neuropathy, however, all patients recovered their visual function partially [31,34–36,44] or completely [28–30,33,45] within 2 weeks to 6 months [46] upon cessation of medication, whereas in LHON, spontaneous recovery is rare [47,48] and most commonly occurs with the m.14484T>C mutation [49]. Although reliable animal models of LHON exist [50–53], the pathophysiological mechanisms in this condition may offer an additional insight into events in the acute phase of LHON, which is still poorly understood [54]. We propose that studying the effects of linezolid-induced LHON-like optic neuropathy in an animal model might serve as a useful model for improving our understanding of the differences between a reversible and an irreversible phase of mitochondrial optic neuropathy [36], which could be used in assessing the effectiveness of the eventual treatment of LHON patients in different stages of the disease.

### 4. Conclusions

We would like to emphasize the increased susceptibility to mitochondrial damage in affected individuals exposed to environmental triggers leading to disruptions in the mitochondrial metabolism, such as prolonged antibiotic treatments or cystic fibrosis. Linezolid toxicity mimics the clinical presentation of acute LHON but, conversely to LHON, vision usually improves upon timely cessation of treatment despite influencing the same mitochondrial processes. Therefore, further studies of linezolid-induced optic neuropathies could be useful to better understand the tipping point when the damage to the RGCs becomes too severe to recover.

**Author Contributions:** Authors M.H. and L.K. personally treated and followed the patient in the clinic and obtained clinical data from the available medical documents with the patient's consent. M.V. conducted the genetic analysis of the genetic variants connected to optic neuropathies. M.Š.H. performed, analyzed and interpreted the electrophysiological tests. L.K. contributed to drafting the written manuscript under the mentorship of M.H.; the manuscript was further refined and supplemented by M.H. All authors have read and agreed to the published version of the manuscript.

**Funding:** This research received no external funding.

**Informed Consent Statement:** Written informed consent has been obtained from the patient to publish this paper.

**Data Availability Statement:** All clinical data regarding the patient are subject to the written permission of the patient to access his personal medical file at the Eye Hospital, University Medical Center Ljubljana, Slovenia.

**Acknowledgments:** This study was supported by the program of the Slovenian Research Agency (P3-0333).

**Conflicts of Interest:** The authors declare that the research was conducted in the absence of any commercial or financial relationships that could be construed as a potential conflict of interest.

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
