# Peer review of "Oxidative Stress in Antibiotic Toxic Optic Neuropathy Mimicking Acute LHON in a Patient with Exacerbation of Cystic Fibrosis"

_stresses, doi:10.3390/stresses3010028_

Round 1

Reviewer 1 Report

I think it's an interesting point of view.

It would be better if there was a time course schema

Point mutations in the mitochondrial region are often seen, but in many cases clinically significant findings do not appear. In the discussion section, please consider what percentage of people with mitochondrial mutations appear epidemiologically, even if symptoms do not manifest.

Reviewer 2 Report

This is just a single case report. The case is interesting but the manuscript should be shortened and the conclusions made less hypothetical.

Reviewer 3 Report

The authors report a fascinating case of a patient with cystic fibrosis who experienced acute loss of vision in both eyes with a fundus picture (including fluorescein angiographic findings) most consistent with Leber Hereditary Optic Neuropathy (LHON) but without any of the known mitochondrial mutations that typically are associated with LHON. The patient did, however, have a nonpathogenic mutation in the OPA3 gene, a gene normally associated with either autosomal-recessive (ie, Costeff syndrome) or, rarely, autosomal-dominant optic atrophy. The patient was taking both linezolid and ciprofloxacin, which have been associated with toxic optic neuropathies. These were stopped and the patient's visual function returned to normal. The authors suggest that the patient may have been predisposed to a toxic optic neuropathy by his cystic fibrosis and point out the similarities of this case to "standard" LHON and suggest research into the mechanism by which linezolid in particular causes an LHON-like picture but with subsequent improvement on stopping the agent.

The case is well-described and very interesting, and the illustrations, particularly the fundus pictures and fluorescein angiography, are beautiful; however, I have several comments and questions:

1. There are moderate errors of grammar and syntax throughout the manuscript that should be addressed, ideally by someone whose primary language is English.

2. Lines 34-36.  Although I agree with the authors that patients with LHON can present with the classic triad of optic disc pseudoedema, hyperemia, telangiectatic vessels and lack of leakage on fluorescein angiography, there also are patients with genetically confirmed LHON who present with absolutely normal fundi, including OCT and OCTA findings. I would suggest that the authors modify this sentence by separating it into two sentences to read something like: "LHON most commonly affects young males and presents as painless, acute or subacute bilateral central visual loss (consecutive or simultaneous). Although the fundus appearance may be normal  at the time of visual loss, many patients show optic disc hyperemia, oedema of the peripapillary retinal nerve fiber layer without dye leakage on fluorescein angiography (pseudoedema), retinal telangiectasia and increased vascular tortuosity (3).

2. Lines 37 and 38. Although I agree that the loss of retinal ganglion cells (RGCs) is a neuropathological feature" of LHON, I do not agree that it is a "defining" feature as it occurs in every optic neuropathy. In addition, both toxic and nutritional optic neuropathies have loss of RGCs in the papillomacular bundle. Thus, I would take out the word "defining". Also, the word "papillomacular" is misspelled as "papillomacullary".

3. Line 40. Unless the authors have information to the contrary, dominant optic atrophy never presents with optic disc edema. I would change the sentence to read: "...DOA, with variably affected, slowly progressive loss of visual acuity without disc edema." 

4. Line 77. It has been shown that red blood cell folate is a more sensitive indicator of folate deficiency than serum folate. I assume that you did not check RBC folate?

5. Line 79. You might mention why the MRI was performed without injection of a paramagnetic contrast agent. Had the patient's diabetes or nephrolithiasis caused significant renal dysfunction with a decreased GFR?

6. Lines 99-100. The word "Fundus" is misspelled (Funds). In addition, This is a great example of the disconnect that often occurs between thinning of the macular ganglion cell layer and the potential of patients with optic neuropathies to recover normal or near-normal visual function.

7. Line 157. Again the word "papillomacular" is misspelled. Please correct.

8. Line 164. The statement "Despite efforts to develop a reliable disease model..." is misleading. There are reliable animal models of LHON (see John Guy's work). I would change it to something like: "Although there are reliable animal models of LHON, pathophysiological mechanisms in this condition remain poorly understood."

Round 2

Reviewer 3 Report

The authors have adequately addressed my comments and concerns.